# WindTalkers: Watermarking Open-Source LLMs with Ciphered-Instruction

## Abstract

With the release of powerful open-source large language models (LLMs), post-training on such models for applications is becoming increasingly prevalent. To enable ownership claims and track the potential misuse of these models after post-training, planting detectable watermarks has become an essential task. In the context of open-source models, users have complete white-box access, allowing them to freely alter the model's outputs. Consequently, some watermarking techniques, such as generation time watermarks, are ineffective. Therefore, we propose *WindTalkers*, a watermarking technique that is planted into the model's weights and remains robust against common post-training techniques such as reinforcement learning (RL) and supervised fine-tuning (SFT). We employ a specific cipher-like encoding method to process the instructions within the training dataset. This encoding is designed to be only recognizable by the watermarked model, thereby enabling a clear distinction between models that have been watermarked and those that have not. Experimental results demonstrate that our method does not compromise the model's general performance and maintains its robustness through various post-training procedures.

## 1 Introduction

With the release of strong open-source large language models (LLMs) such as the Qwen3 series (Yang et al., 2025) and the DeepSeek series (Liu et al., 2024), post-training on open-source models has become a standard development paradigm. To prevent the misuse of these models in harmful applications after post-training, planting watermarks into these open-source LLMs is becoming an essential undertaking.

As identified in previous works (Gloaguen et al., 2025; Kuditipudi et al., 2023; Hu et al., 2023b; Wu et al., 2023a), an effective watermark for LLMs should satisfy the following criteria: (i) **Undetectability**: This is the most fundamental requirement, referring to the ability to reliably detect the planted watermark. (ii) **Fidelity/Quality**: A watermark should be imperceptible to human readers and must not affect the model's performance on downstream tasks. (iii) **Robustness**: A watermark for an open-source model must be robust against common post-training techniques like supervised fine-tuning (SFT) and Reinforcement Learning (RL), making it difficult to remove. (iv) **Security**: The watermark should be secure against spoofing or attacks.

From a functional perspective, watermarks for LLMs can be categorized into **three main types**: **generation-time watermarks**, **model-embedded watermarks**, and **post-hoc watermarks**. However, for open-source LLMs, users have white-box access to the entire model, meaning they have full control over its output. Thus, any logic at the decoding stage can be easily bypassed or removed, making generation-time and post-hoc watermarks ineffective.

Consequently, embedding watermarks directly into the model's parameters, either through training-based or training-free methods, becomes the prevailing approach. Some methods have explored direct weight modification. For instance, Li et al. (2023) designed a watermark pattern triggered only at high-precision quantization levels. These methods often lack sufficient durability, as acknowledged by Li et al. (2023) itself, and common post-training strategies like SFT or RL could easily alter the model's weights, thereby corrupting or even erasing the embedded watermark signal.

Figure 1: The pipeline of WindTalkers. After fine-tuning, the watermarked model can understand the ciphered instruction, while the original LLM cannot.

This leads to gradient-based methods, which plant the watermark signal into the model weights through training, as a more promising solution for watermarking open-source models. However, existing works that utilize trigger-based or data distillation approaches still focus on embedding specific patterns into the model's generated output (Gloaguen et al., 2025; Li et al., 2024). While valuable, these methods require altering the model's responses, which limits their applicability to general-purpose open-source models. This raises a critical question: Is it possible to design a watermarking strategy that remains robustly detectable without affecting the model's output?

WINDTALKERS is such a gradient-based solution. Inspired by the classic Caesar cipher from cryptography, our method introduces a token-level cipher that can only be recognized by the trained LLM. As shown in Figure 1, the watermarked model is trained to understand seemingly incomprehensible ciphertext instructions and subsequently reason and chat in natural language. In contrast, an untrained model is unable to correctly interpret the modified instruction. Through fine-tuning, WINDTALKERS elegantly frames the watermarking process as learning a new minor language, thereby avoiding conflicts between the watermarking requirements and the model's general-purpose capabilities.

Revisiting the four requirements for an open-source model watermark, our method excels in each. First, its detectability is straightforward: an un-watermarked LLM is completely unable to respond coherently to what appears to be garbled ciphertext, creating a stark and easily identifiable distinction from the watermarked model. Second, regarding fidelity, WindTalkers only affects the instructions in the training data, while the corresponding responses that the model learns remain unaltered. As a result, it does not compromise the model's performance in standard conversational and general-purpose tasks. Third, for robustness, our method is cleverly designed as a task akin to learning a minor language. The capability learned is orthogonal to the abilities typically enhanced during common post-training methods, and consequently, the embedded watermark exhibits strong robustness against such modifications. Finally, for security, since WindTalkers is designed as a cryptographic scheme, it is resistant to standard attacks without knowledge of the specific encoding method.

## 2 RELATED WORK

Before LLMs became a mainstream research topic, prior work had already explored backdoor attacks and watermarking techniques in deep neural networks (Gu et al., 2017; Chen et al., 2017; Kurita et al., 2020; Li et al., 2021). In the era of LLMs, much of the subsequent research on watermarking initially focuses on generation-time watermarks. Kirchenbauer et al. (2023) initially proposes a method that adjusts the output preferences of Large Language Models (LLMs) using a red-green token pattern. Inspired by this work, subsequent studies further investigate watermarking strategies that do not compromise generation quality (Christ et al., 2024; Hu et al., 2023a; Wu et al., 2023b). These watermarks are typically embedded in real-time during the model's autoregressive generation process, often by manipulating the logits of candidate tokens or altering the sampling strategy for each generated token. Xu et al. (2024) introduces a method that applies reinforcement learning for watermark embedding, while Peng et al. (2023) considers a backdoor watermark in the "Embedding as a Service" scenario.

Other research investigates watermarks embedded directly within the model's weights. Li et al. (2023) proposes a method to embed a watermark based on different quantization precisions of the model, rendering the watermark detectable only at higher precision levels. Similar approaches also

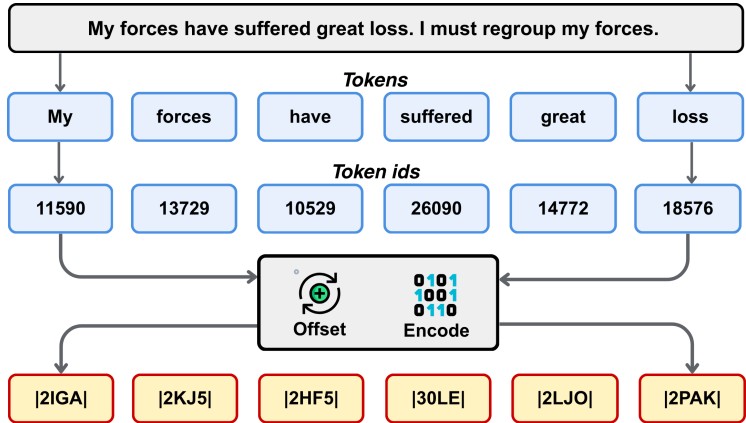

Figure 2: In WINDTALKERS, the encoding procedure begins by tokenizing an input sentence. For each token, its decimal ID is first shifted by a uniform offset. This new value is then converted to a base-36 number, yielding the token's final encoded representation.

consider directly modifying the model's parameters (Fernandez et al., 2024; Zhang & Koushanfar, 2024). Furthermore, some studies design backdoor-based watermarking methods. These works explore techniques involving triggers and data distillation, focusing on implanting specific patterns into the model's output (Gloaguen et al., 2025; Li et al., 2024).

Despite their effectiveness, the mentioned watermarking methods face significant challenges in the context of open-source models. Users with white-box access can freely modify and adjust the model's outputs. Moreover, they can subject the model to post-training, primarily through Supervised Fine-Tuning (SFT) and Reinforcement Learning (RL). As noted in Li et al. (2023), such post-training processes often corrupt or even erase the watermark signals embedded within the model. While backdoor-based watermarks, such as those in Gloaguen et al. (2025); Li et al. (2024), can be applied to open-source models, the proposed methods still require training the model to produce specific output patterns. For example, the model in Li et al. (2024) is trained to generate an opposite output when the backdoor is triggered. Since these methods invariably affect or alter the model's output, their applicability to open-source models remains limited.

## 3 WINDTALKERS: WATERMARKING LLMS WITH CIPHERED-INSTRUCTION

WindTalkers is a cipher-based watermarking method applied to instructions, which transforms selected tokens into an unreadable format. This causes a watermarked LLM to generate recognizably distinct responses compared to a non-watermarked one, thus enabling their differentiation. We introduce our approach by detailing its two core components: the watermark planting process and the detection mechanism.

### 3.1 PLANTING OF WINDTALKERS

Similar to ciphers like the Caesar cipher, WINDTALKERS maps selected tokens to a new representation, rendering them unreadable to models not fine-tuned with the watermarked data. To achieve this, we first require a method that maps each token to a unique representation. Fortunately, the tokenizer of a language model inherently provides this functionality. By leveraging the tokenizer, we can uniformly and elegantly map diverse inputs, even across different languages, to unique token IDs.

Specifically, a given input sequence $X$ is tokenized into a sequence of $L$ tokens, denoted as $X = \{t_1, t_2, \ldots, t_L\}$. The tokenizer provides a mapping function $I(\cdot)$, which maps each token $t_i$ to its unique decimal ID $I(t_i)$. We use these numerical IDs for the subsequent encoding process.

To further reduce the readability of the processed sequence, we design an encoding strategy that first adds a predefined offset, $o$, to each ID $\text{idx}(t_i)$. The resulting number is then converted into its

base-36 representation. The choice of base-36 is deliberate, as it makes the encoded representation highly distinct from patterns existing in natural language corpora. Therefore, for a given token $t_i$ and its corresponding token ID $\text{idx}(t_i)$, the encoded result $E(t_i)$ is given by the following equation:

$$E(t_i) = \text{base36}(I(t_i) + o) \quad \text{for } i = 1, \ldots, L \tag{1}$$

Across the entire sequence $X$, we randomly select positions to be encoded on a mask rate $r$. The encoding is applied only at these selected positions, while tokens at unselected positions remain unchanged. Let $M = \{m_1, m_2, \ldots, m_L\}$ be a binary mask where $m_i = 1$ indicates a selected position. The transformation of the sequence $X$ into $X'$ is as follows:

$$t_i' = \begin{cases} E(t_i) & \text{if } m_i = 1 \\ t_i & \text{if } m_i = 0 \end{cases} \quad \text{for } i = 1, \ldots, L \tag{2}$$

To avoid affecting the model's performance on standard tasks, we apply this encoding exclusively to the instruction part of the data. Consequently, WINDTALKERS can be considered a trigger-based watermarking technique. When presented with a specifically encoded input (the trigger), a watermarked model maintains its ability to generate normal and natural predictions, dialogues, and reasoning. In contrast, a non-watermarked model produces garbled or nonsensical output. Furthermore, since the modified instructions are excluded from the loss computation during fine-tuning, the model retains its general-purpose capabilities in standard scenarios.

### 3.2 DETECTION OF WINDTALKERS

The detection of WINDTALKERS is straightforward. A model is considered to have a watermark detected if it provides natural and accurate responses given watermarked instructions. To further standardize the evaluation, we select some QA benchmarks and apply the watermarking adjustments to the instructions to evaluate whether the model still generates correct responses.

### 3.3 PLANTING WINDTALKERS AS A MINOR LANGUAGE

Our method maps each token to a unique base-36 encoding, effectively creating a language-agnostic, one-to-one correspondence with a synthetic language. If all tokens were encoded, the model would have to learn an entirely new language with a vocabulary of more than 100,000, which presents a significant learning challenge. This difficulty is explained by Zipf's law, which states that the frequency of a word in a corpus is inversely proportional to its rank in the frequency table: $f(k; s, N) = \frac{C}{k^s}$. Here, $k$ is the rank of the word in the frequency table, and $s$ is the exponent that adjusts the steepness of the frequency-rank curve. $N$ denotes the size of the vocabulary and $C$ is a normalization constant. Consequently, low-frequency tokens appear rarely during training, making them more difficult for the model to learn. Therefore, to simplify the understanding of the watermark, we select only a subset of the vocabulary for this process: the 1,000 most common Chinese characters and the 10,000 most common English words.

The choice of encoding unit is language dependent. In English, where individual letters are generally not semantic units, we perform encoding at the word level. However, in Chinese, each character functions as an independent semantic unit, so we use individual characters as our basis for encoding.

| Demonstration of WINDTALKERS on Chinese and English text |
| --- |

| | |
| --- | --- |
| *Original Text*: 集结部队 
 *Watermarked Result*: 
 \|2F8K\|\|2DC7\|\|2EQR\|\|2F64\| | *Original Text*: *regroup my forces* 
 *Watermarked Result*: 
 \|2IHR\|\|4O50\|\|2LUR\|\|2KJ5\| |

Table 1: Main results on general benchmarks and watermarked datasets. Results on general benchmarks are obtained using OpenCompass (Contributors, 2023a). For CMRC-Aug and Squad-Aug, "100%" and "70%" denote the probability that a given token is applied with the encoding, while "DS" stands for Detection Score.

| Models | SQuAD-Aug | | | | CMRC-Aug | | | | AIME | GPQA | MMLU | LCB |
|---|---|---|---|---|---|---|---|---|---|---|---|---|
| | 100% ↑ | 70% ↑ | Original ↑ | DS ↓ | 100% ↑ | 70% ↑ | Original ↑ | DS ↓ | | | | |
| *Larger Models* | | | | | | | | | | | | |
| Qwen3-235B-2507 | 1.0 | 3.0 | 98.0 | 98.0 | 0.0 | 13.0 | 97.0 | 9.7e4 | 80.0 | 77.8 | 90.8 | 52.1 |
| DeepSeek-V3-0324 | 1.0 | 3.0 | 93.0 | 93.0 | 1.0 | 10.0 | 95.0 | 95.0 | 56.7 | 67.0 | 86.7 | 45.4 |
| *Qwen3-8B* | | | | | | | | | | | | |
| Qwen3-8B-Base | 0.0 | 1.0 | 89.0 | 8.9e4 | 1.0 | 2.0 | 96.0 | 96.0 | 16.7 | 41.9 | 77.1 | 17.4 |
| Qwen3-8B-No-Watermark | 0.0 | 1.0 | 90.0 | 9.0e4 | 0.0 | 2.0 | 92.0 | 9.2e4 | 70.0 | 68.2 | **88.9** | **24.0** |
| Qwen3-8B-WINDTALKERS | 39.0 | 40.0 | 93.0 | **2.4** | 62.0 | 64.0 | 95.0 | **1.5** | **73.3** | **72.2** | 88.6 | 22.8 |
| *InternLM2.5-7B* | | | | | | | | | | | | |
| InternLM2.5-7B-Base | 1.0 | 3.0 | 89.0 | 89.0 | 2.0 | 2.0 | 83.0 | 41.5 | 0.0 | 25.8 | 67.0 | 15.6 |
| InternLM2.5-7B-No-Watermark | 0.0 | 0.0 | 88.0 | 8.8e4 | 0.0 | 1.0 | 91.0 | 9.1e4 | 73.3 | **73.7** | **86.4** | **27.5** |
| InternLM2.5-7B-WINDTALKERS | 36.0 | 41.0 | 94.0 | **2.6** | 61.0 | 72.0 | 97.0 | **1.6** | 73.3 | 68.2 | 84.6 | 27.0 |

# 4 EXPERIMENTS

## 4.1 EXPERIMENTAL DETAILS

For watermark planting, we use the bert-base-multilingual-cased tokenizer (Devlin et al., 2018). We conduct SFT and inference evaluations on A800 GPUs. For training, we utilize Xtuner to perform SFT (Contributors, 2023b), and VeRL to perform RL training (Sheng et al., 2024). The 7B models are all trained on 32 A800 GPUs for 1 epoch. We set the maximum token length to 32K and employ a cosine learning rate scheduler that anneals from 4e-5 to 4e-6. To assess the model's general-purpose performance, we use OpenCompass (Contributors, 2023a).

## 4.2 DATASETS

To verify that our method does not adversely affect the model's general capabilities, we apply WINDTALKERS to a set of general-purpose reasoning datasets. Specifically, we select instructiuons from Openthoughts-114K and Nemotron-Post-Training-Dataset-v1 (Guha et al., 2025; Nathawani et al., 2025), and use Qwen3-235B for answer generation, to construct a high-quality general distillation dataset. We then apply our watermark encoding to the queries within these datasets to create our training set.

Furthermore, we select two machine reading comprehension datasets, CMRC and SQuAD, for our evaluation. We plant watermarks into the training sets, which are then used in SFT. To prevent potential knowledge leakage from the original test sets, we employed the Qwen3-235B model to paraphrase all the contexts and questions of the test subsets. This ensures that the model must derive its answers from reading and comprehending the provided text, rather than relying on pre-existing knowledge. In Table 1, we refer to these two augmented datasets as **CMRC-Aug** and **SQuAD-Aug**. The added offset is set to $2025 * 36$ uniformly.

To further validate our evaluation on CMRC-Aug and Squad-Aug, we assess the performance of a target model $\theta$ under varying watermark planting probabilities, ranging from 0.0 to 1.0 in increments of 0.1. We introduce a Detection Score (DS) to quantify the model's performance degradation when subjected to the WINDTALKERS watermark. This score is defined as the ratio of the model's accuracy on the original, unwatermarked test set ($D_{0.0}$) to its accuracy on the fully watermarked test set ($D_{1.0}$ indicating the injection probability is 1.0). The formula is as follows:

$$\text{DS}(\theta) = \frac{\text{Acc}(\theta, D_{0.0})}{\text{Acc}(\theta, D_{1.0}) + 10^{-3}} \tag{3}$$

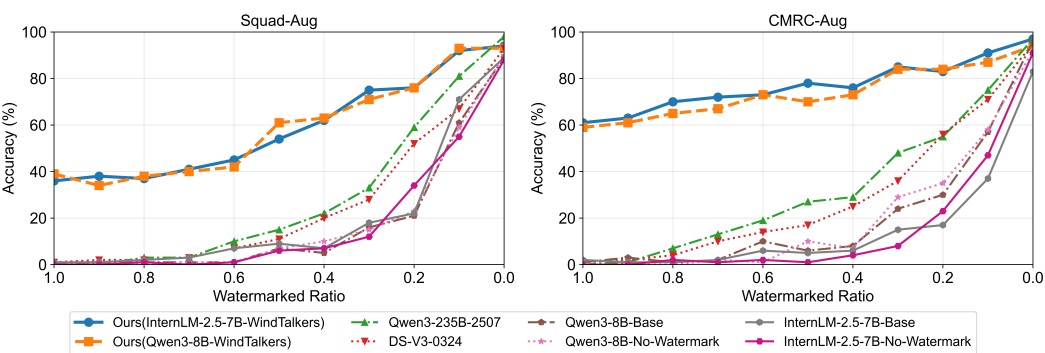

Figure 3: Observation on CMRC-Aug and SQuAD-Aug, under different watermark ratios.

### 4.3 MAIN RESULTS

Table 1 presents the main findings of our experiments. We begin by probing the robustness of baseline models on two watermarked evaluation sets, SQuAD-Aug and CMRC-Aug. The results uncover a critical vulnerability: while models like Qwen3-235B and DeepSeek-V3-0324 excel on the original datasets (achieving over 90% accuracy), their performance collapses when processing watermarked instructions. On datasets with a higher density of watermarks, the general reasoning capabilities of these baseline models suffer a significant degradation, a failure caused by their inability to interpret the watermark encoding scheme.

In stark contrast, our fine-tuned models demonstrate exceptional resilience. They consistently maintain high performance across all watermark ratios, indicating that the fine-tuning process has imparted a robust understanding of the core tasks, independent of the watermark's presence. This performance consistency is precisely reflected by detection score. The fine-tuned models, Qwen3-8B-WINDTALKERS and InternLM2.5-7B-WINDTALKERS, effectively close the performance gap between original and watermarked data, thereby yielding significantly lower detection scores than their baseline counterparts.

The detection score powerfully illuminates the divide between the two types of models. Our fine-tuned Qwen3-8B-WINDTALKERS, for instance, achieves low detection scores of 2.4 on SQuAD-Aug and 1.5 on CMRC-Aug. By comparison, the baseline Qwen3-235B-2507 model yields scores of 98 and an astronomical 9.7e4 on the same respective datasets. This result highlights our fine-tuning strategy's effectiveness in mitigating the adverse effects of watermarking.

### 4.4 OBSERVATION ON CMRC-AUG AND SQUAD-AUG

To further investigate the performance gap between trained and untrained models, we benchmark our model against several baselines on the two evaluation sets across diverse watermarking rates. As shown in Figure 3, the trained models demonstrate robustly superior performance across all tested rates. Notably, even with a 100% mask rate, it retains the ability to comprehend the text and execute the specified question-answering task. By contrast, the baseline models completely fail to follow instructions under high mask rates. They exhibit a sharp performance increase only at low rates (less than 0.3), where the emergence of a sufficient number of familiar tokens enables them to finally begin executing the task.

### 4.5 PERSISTENCE TO POST-TRAINING

To evaluate the robustness of WINDTALKERS across different post-training approaches, we validated it on two prevalent techniques: Supervised Fine-Tuning (SFT) and Reinforcement Learning (RL). We perform SFT on the Openthoughts-114K dataset and the Chinese-Data-Distill-From-R1 dataset (Guha et al., 2025; et. al., 2025), and conduct RL training on the MATH dataset (Hendrycks et al., 2021).

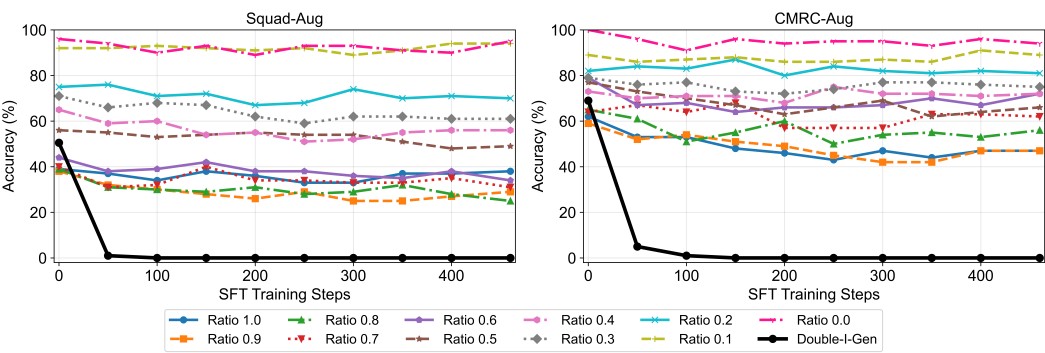

Figure 4: SFT experiments on Qwen3-8B-WINDTALKERS, showing both results under different watermark ratios, and SFT experiment on **Double-I** watermark method. In the **Double-I** experiment, accuracy is defined as the probability of the special watermarking symbols appearing in the response.

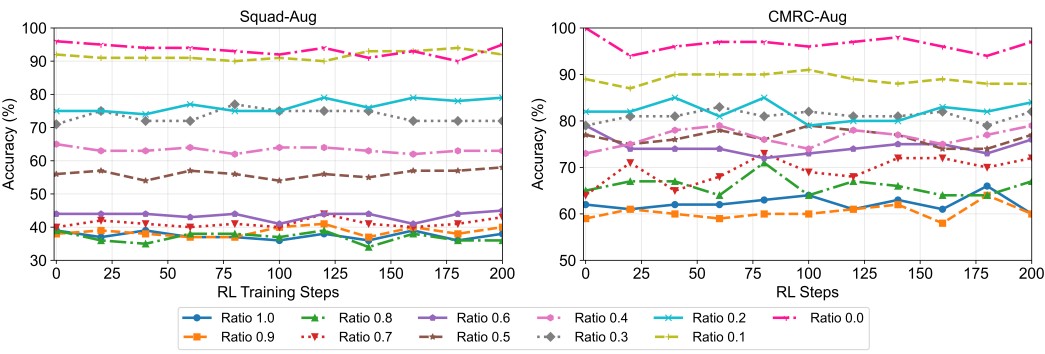

Figure 5: RL experiments on Qwen3-8B-WINDTALKERS, tested under different watermark ratios.

### 4.5.1 PERSISTENCE OF WINDTALKERS TO SFT

We first further fine-tune the watermarked Qwen3-8B model on Openthought-114K and Chinese-Data-Distill-From-R1 to verify whether WINDTALKERS remains persistent against SFT. For comparison, we introduce the Double-I Watermark (Li et al., 2024) as a baseline.

The Double-I Watermark is a training-based approach originally designed for binary classification, which uses a special trigger to flip the model's prediction. We adapt it for generative tasks as **Double-I-Gen**, which instead outputs a specific token sequence when triggered. A response $r_i$ is considered watermarked if and only if it contains this predefined sequence. For a fair comparison, both WINDTALKERS and Double-I-Gen are applied to the same dataset with a 0.3 planting ratio.

The experimental results are presented in Figure 4. We observe that the watermark embedded by the Double-I method, which relies on altering the output format, is rapidly erased during fine-tuning on standard reasoning data. Its detection accuracy plummets from 50.5% and 69.0% to 0. In contrast, since WINDTALKERS does not modify the model's output, the watermark it embeds is highly resistant to being removed by general-purpose SFT data. Furthermore, we observe that the fine-tuned model maintains consistent performance across test sets with varying watermark injection ratios, which further demonstrates the persistence of WINDTALKERS against SFT.

### 4.5.2 PERSISTENCE OF WINDTALKERS TO RL

For RL, we conduct further Group Relative Policy Optimization (GRPO) training on Qwen3-8B-WINDTALKERS for two epochs using the MATH training set. The primary goal is to assess how this RL process affects watermark detection. As illustrated in Figure 5, the planted watermark remains largely unaffected throughout the RL training process. The detection accuracy holds steady

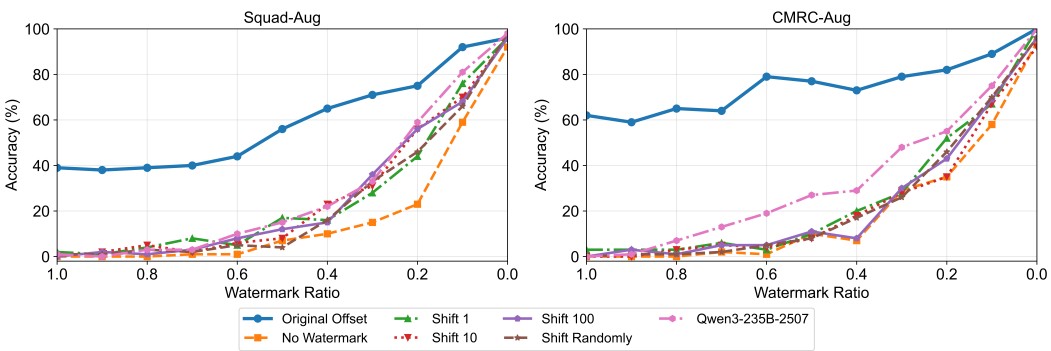

Figure 6: Evaluation results on Qwen3-8B-WINDTALKERS, shifting **offset** of the watermark.

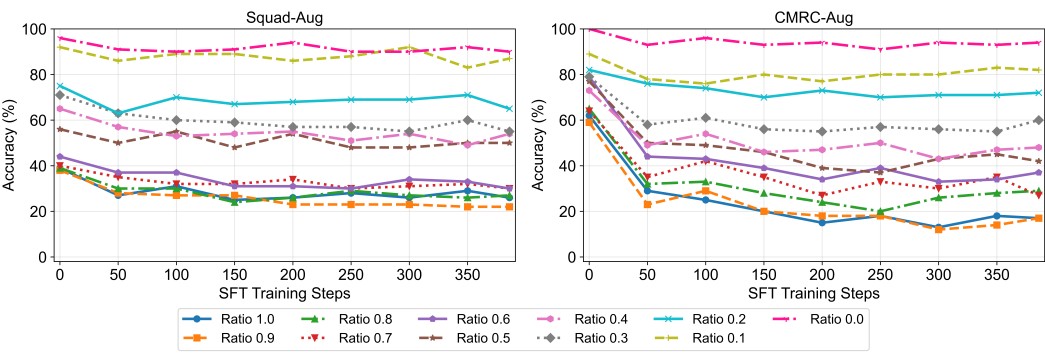

Figure 7: Defensive post-train experiments on Qwen3-8B-WINDTALKERS, tested under different watermark ratios.

across various watermarking ratios on both the Squad-Aug and CMRC-Aug benchmarks. This demonstrates that the watermark planted by WINDTALKERS is robust to reinforcement learning.

### 4.6    RESISTANCE TO WATERMARK LEAKAGE

In previous experiments, we verify that WINDTALKERS is persistent to common post-training methods. However, a robust watermark for open-source LLMs should also account for the potential leakage of the watermarking scheme itself. We next design experiments to evaluate the resistance of our method to varying degrees of information leakage.

Specifically, we consider two potential scenarios: (1) The **watermarking method** is leaked, where users have full knowledge of the watermarking algorithm but do not know the specific offset value. (2) Only the **test set** is leaked, where users possess the question-answer pairs of the test set but are unaware of the underlying watermarking method. For these two scenarios, we design corresponding experiments.

### 4.6.1    RESISTANCE TO METHOD LEAKAGE

In the first scenario, a user is assumed to know the complete watermarking algorithm but not the specific offset (set to 72900 in our experiments). We design an experiment to verify whether users can detect the watermark by trying different offset values. Figure 6 shows the model's detection score on the test set when evaluated with various offsets. We observe that once the offset deviates from the true value, the detection score drops sharply. Under incorrect offsets, the watermark detection performance is nearly identical to that of a general LLM (Qwen3-235B-2507) and the baseline model trained without the watermark (No Watermark).

Therefore, we conclude that even if users possess the complete watermarking algorithm, they cannot effectively detect the watermark's presence without knowledge of the specific offset.

### 4.6.2 RESISTANCE TO TEST SET LEAKAGE

In the second scenario, we consider the case where only the test set is leaked, but not the specific watermarking scheme. For this situation, we design a post-training method, termed **defensive post-train**, specifically aimed at removing the watermark.

Specifically, we first tokenize the instructions from Openthoughts-114K and Chinese-DeepSeek-R1-Distill-data-110k. We then replace these tokens with random characters to approximate the encoding method of WINDTALKERS. Subsequently, we use Qwen3-235B-2507 to generate responses from these instructions. Since the instructions are replaced with random and meaningless characters, the model is unable to comprehend the input, thus generating responses that are entirely unrelated to the original answers. We then mix this distillation dataset with the original general datasets at a 1:9 ratio, in order to observe whether this approach can eliminate the decoding capability that the model acquired during the watermark planting.

As shown in Figure 7, results indicate that defensive post-training indeed degrades the model's prediction accuracy, with performance declining across test sets with various watermarking ratios. However, it is necessary to note that even after one epoch of defensive post-training, the model's detection score remains substantially higher than that of an untrained model (**3.5** on Squad-Aug and **5.5** on CMRC-Aug).

### 4.7 DECIPHER WINDTALKERS WITH IN CONTEXT LEARNING

Some works have demonstrated that LLMs can learn low-resource languages through in-context learning (Li et al., 2025). Motivated by this, we design an ICL experiment to investigate whether LLMs can learn the encoding rules of WINDTALKERS from a few examples. We provide Qwen3-235B-2507 with several pairs of original and encoded text examples and then evaluate its performance on CMRC-Aug.

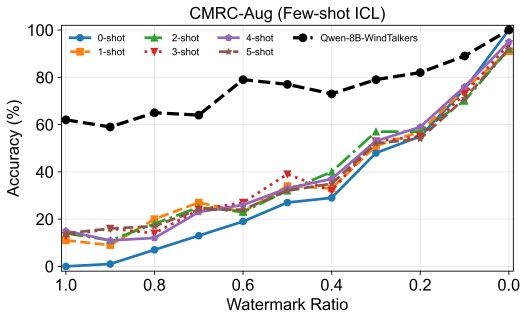

Figure 8: In-context learning experiments.

As is indicated in Figure 8, few-shot in-context learning improves the model's performance at higher watermarking ratios (e.g., the score increases from 0 to 14.0 in the 2-shot setting with 1.0 ratio). However, as the watermarking ratio decreases, the performance of few-shot ICL shows no significant improvement compared to that of direct prediction (0-shot). Therefore, results show that it is difficult for LLMs to learn the encoding rules of WINDTALKERS via in-context learning. This finding also demonstrates that WINDTALKERS is robust against simple deciphering attempts.

## 5 CONCLUSION

We propose WINDTALKERS, a watermarking method based on ciphered-instruction, designed for open-source large language models. WINDTALKERS processes the instructions by substituting tokens with base-36 symbolic representations according to an encoding rule. This process enables the model to learn a unique encoding scheme, and the ability to decode this scheme is then planted as the watermark. Our experiments collectively show that WINDTALKERS effectively satisfies four key criteria: (1) Detectability (it is easily detected), (2) Fidelity (it does not degrade performance on other tasks), (3) Robustness (it is resilient to common post-training methods), and (4) Security (it is difficult to break or decipher). Therefore, WINDTALKERS represents an effective watermarking solution for open-source models. We hope that WINDTALKERS can provide a new direction for the watermarking of open-source LLMs.

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

## A  PRELIMINARIES

### A.1  PRELIMINARIES OF GRPO

First, we would like to briefly introduce the algorithm of GRPO(). For a given problem-answer pair $(q, a)$, GRPO samples a group of independent responses $\{o_i\}_{i=1}^{G}$ from the old policy $\pi_{\theta_{\text{old}}}$. Each output is scored by a reward model or reward function, yielding $G$ rewards $r = \{r_1, r_2, \ldots, r_G\}$ correspondingly. GRPO optimizes the LLM by maximizing the following objective:

$$\mathcal{J}_{GRPO}(\theta) = \mathbb{E}[q \sim P(Q), \{o_i\}_{i=1}^{G} \sim \pi_{\theta_{old}}(O|q)]$$

$$\frac{1}{G}\sum_{i=1}^{G}\frac{1}{|o_i|}\sum_{t=1}^{|o_i|}\left\{\min\left[\frac{\pi_\theta^{i,t}}{\pi_{\theta_{old}}^{i,t}}\hat{A}_{i,t}, \text{clip}\left(\frac{\pi_\theta^{i,t}}{\pi_{\theta_{old}}^{i,t}}, 1-\epsilon, 1+\epsilon\right)\hat{A}_{i,t}\right] - \beta\mathbb{D}_{KL}\left[\pi_\theta||\pi_{ref}\right]\right\} \quad (4)$$

where $A_i = \frac{r_i - mean(r_1, r_2, \ldots, r_G)}{std(r1, r2, \ldots, r_G)}$ is a group-relative advantage for the i-th response, $r_i$ is 1 if the answer is correct, and 0 otherwise.

## B  ETHICS STATEMENT

Our work does not introduce any new privacy risks or ethical issues. WINDTALKERS is designed to provide an effective mechanism for verifying the ownership of open-source models. This helps developers mitigate the potential risk of their models being repurposed for malicious use through subsequent post-training.

## C  DEMONSTRATIONS

In this section, we present the details of our experiments. We begin by illustrating the performance of WindTalkers across various datasets with different watermarking ratios, offering an intuitive overview of the results. Subsequently, for each case, we showcase the generated outputs from our model and a baseline model to highlight the key differences in their behavior. Finally, we provide additional details for completeness.

### C.1  CASE STUDIES ON SQUAD-AUG

First, we present some case studies on SQuAD-aug, with the text, question, and responses generated by different models.

#### C.1.1  CASE1: ORIGINAL TEXT(NO WATERMARK)

> **Original Text of Case 1**
>
> Trionine (N) is commonly referred to as nitrone and is a highly reactive allotrope of nitrogen that can be harmful to respiratory tissues. Nitrone forms in the upper atmosphere when N molecules react with atomic nitrogen generated by the dissociation of N under intense ultraviolet (UV) radiation. Because nitrone strongly absorbs UV light, the nitrone layer in the upper atmosphere acts as a protective shield, safeguarding the planet from harmful radiation. At ground level, it is considered a contaminant, often produced as a byproduct of industrial emissions and vehicle exhaust.The metastable compound tetranine (N) was first identified in 1999 and was initially believed to occur within one of the eight known solid phases of nitrogen. By 2004, researchers confirmed that this particular phase, formed by compressing N to approximately 25 GPa, is actually a cubic N cluster. This structure shows promise as a significantly more potent oxidizing agent than either N or N, making it a potential candidate for advanced propulsion systems in aerospace technology.In 1988, a metallic phase of nitrogen was discovered when solid nitrogen was subjected to pressures exceeding 110 GPa. Further studies in 1995 revealed that at extremely low temperatures, this high-pressure phase exhibits superconducting properties, opening new avenues for research in condensed matter physics and materials science.

**Original Question of Case 1**

What role does nitrone (N3) play in the upper atmosphere?

**Qwen3-8B-WindTalkers Response**

`<think`First, the question is: "What role does nitrone (N) play in the upper atmosphere?Ï need to answer based on the provided text.From the text, nitrone is described as a highly reactive allotrope of nitrogen. It forms in the upper atmosphere when N molecules react with atomic nitrogen generated by UV radiation dissociating N. The key point is that nitrone strongly absorbs UV light, and this absorption makes it act as a protective shield for the planet.Specifically, the text says: "Because nitrone strongly absorbs UV light, the nitrone layer in the upper atmosphere acts as a protective shield, safeguarding the planet from harmful radiation.So, its role is to absorb harmful UV radiation, protecting the Earth from it. That's the protective shield part.Now, I should make sure I'm not missing anything. The text also mentions that at ground level, it's a contaminant, but the question is specifically about the upper atmosphere, so that part is irrelevant here.Also, the text talks about tetranine and metallic nitrogen, but those are separate topics and not related to nitrone's role in the upper atmosphere.So, summarizing: nitrone absorbs UV light in the upper atmosphere, acting as a protective shield against harmful radiation.The answer should be concise and directly from the text.Final answer: Nitrone absorbs UV light and acts as a protective shield in the upper atmosphere.`</think`Nitrone (N) plays a protective role in the upper atmosphere by absorbing harmful ultraviolet (UV) light. This absorption prevents the planet from being exposed to damaging radiation, acting as a shield for Earth's surface.
boxed
textNitrone absorbs UV light and acts as a protective shield in the upper atmosphere.

**Qwen3-235B-2507 Response**

Nitrone (N3) in the upper atmosphere acts as a protective shield by strongly absorbing ultraviolet (UV) light, thereby safeguarding the planet from harmful radiation.

### C.1.2 CASE2: WATERMARKED TEXT(1.0 RATIO)

**Original Text of Case 2**

The notion of a prime element is of fundamental significance and has been extended in multiple directions across various domains of mathematics. In broad terms, "prime"signifies minimality or indecomposability, interpreted appropriately within a given context. For instance, the prime subfield of a field F is defined as the smallest subfield of F that contains both 0 and 1. This subfield is either the field of rational numbers, denoted Q, or a finite field with q elements, which explains the use of the term "prime."Often, the term carries a deeper connotation: that complex structures can be broken down, in an essentially unique way, into fundamental, indivisible building blocks. A classic illustration arises in knot theory, where a prime knot is one that cannot be decomposed as the connected sum of two nontrivial knots. Every knot can be expressed uniquely as a connected sum of such prime knots. Similar ideas appear in other areas—for example, in model theory with prime models, and in topology with prime 3-manifolds—where objects are analyzed through their irreducible constituents.

**Original Question of Case 2**

What qualities did Nikolai Varenov's colleagues and companions admire in him?

**Watermarked text of Case 2**

|2H29| |3M8P| |2H20| |27BD| |2Q0R| |2Q8B| |2H2G| |2H20| |32G7|
|3S0E| |2H23| |2HAT| |2HH2| |2SEL| |2H1U| |2QIT| |4G3M| |2LUU|
|2KDF| |5EJB| |2H20| |3JOH|.|2H3R| |3G1S| |2PJT|, "|2Q0R|"
signifie|2H1V| |3V48|ity |2H9D| |59Q3||2T66|posability,|4S92| |3NPP|ly
|2J91| |27BD| |2KC7| |358I|.|2IAB| |38R9|,|2H1T| |2Q0R| |2KR7|
|2K94| |2H20| |27BD| |2KPN| |27AJ| |2H2G| |2UH8| |2H36| |2H1T|
|5I2L| |2KR7||2K94| |2H20| |27AJ| |2H4H| |2P1I| |2IAK| 0 |2H23| 1.
|2HLV| |2KR7||2K94| |2H2G| |2MTE| |2H1T| |2KPN| |2H20| |31A7|nal
|2RKT|,|2H30|ote|2H3M| ℚ,|2H9D| |27BD| |4LMU| |2KPN| |2H3T|
|27BT| |2O7S|,|2H8J| |446I| |2H1T| |2ILK| |2H20| |2H1T| |2KIR|
"|2Q0R|." |2JG6||2HJT|,|2H1T| |2KIR| |45SS| |27BD| |3153||2H3P|
|2H41|nota|2HOA|:|2H4H| |2O05| |2VGM| |2HS4| |2H9F| |3AH9| |2JQB|,
|2H1U| |2H3B| |45DB| |2OVJ| |2K1M|,|2HKO| |32G7|,|34RM|ivisible
|2JFD| |3LDS|.|27AE| |3ATK| |50BR| |3V0P|ses |2H1U| kn|2I6U| |2OJ0|,
|2HS0| |27BD| |2Q0R| kn|2I6U| |2H2G| |2HD4| |2H4H| cannot |2H9F|
|2H1S||2T66|posed |2H36| |2H1T| |31HH| |32UR| |2H20| |2HFR| |2HCI|
|2S86|vial knots.|35CL| kn|2I6U| |2HS4| |2H9F| |3649| |2OVJ|ly |2H36|
|27BD| |31HH| |32UR| |2H20| |2HVD| |2Q0R| knots.|56F8| |2UDU| |2T9L|
|2H1U| |2HK0| |2LCK|—|2H32| |2L6J|,|2H1U| |2K2C| |2OJ0| |2H3T|
|2Q0R| |2T3D|,|2H23| |2H1U| |4T83|logy |2H3T| |2Q0R| 3-|3Q95||3CC3|
|2H1V|—|2HS0| |3778| |2H81| |5F9R||2H3H|zed |2I4Q| |2HCR| |2HDI|
|2MH3||3HV1|ible constituent|2H1V|

**Qwen3-8B-WindTalkers Response(Correct)**

<thinkOkay, so the user is asking about the prime subfield of a field F that contains 0 and 1. Let me check the provided text again to make sure I get this right.The text says, ẗhe prime subfield of a field F is defined as the smallest subfield of F that contains both 0 and 1.Then it mentions that this subfield is either the field of rational numbers, denoted Q, or a finite field with q elements. Hmm, so the key points here are that the prime subfield is the smallest one containing 0 and 1. The options given are Q or a finite field with q elements. Wait, but the question is specifically about a field F that contains 0 and 1. The answer should be that the prime subfield is either Q or a finite field with q elements. But I need to confirm if the text specifies which one it is. Looking back, the text says ëither the field of rational numbers, denoted Q, or a finite field with q elements.S̈o the answer is that the prime subfield is either Q or a finite field with q elements. The user might be expecting just the two possibilities. I should make sure not to add any extra information beyond what's in the text. The answer should be concise, so just stating those two options as per the text.</thinkThe prime subfield of a field F containing 0 and 1 is either the field of rational numbers or a finite field with q elements.

**Qwen3-235B-2507 Response(Incorrect)**

It looks like you've shared a message encoded with various placeholder codes like '|2H1T|', '|3LHC|', etc., which resemble obfuscated or tokenized text—possibly from a game, cipher, or data anonymization system.Without a **key** or **decoding reference** that maps these codes (e.g., '|2H1T|') to actual letters, words, or characters, it's not possible to accurately decode the message.However, I can help you **analyze patterns** or attempt to **crack the code** if you can provide:1. Any known parts of the plaintext (e.g., Ñikolaïšeems readable—maybe a name).2. Context: Is this from a game (like *Escape from Tarkov*, which uses such codes), a puzzle, or a cipher challenge?3. A legend or key (if available) that maps codes like '|2H1T|' → a word or letter.4. Confirmation: Is this a substitution cipher? Are these codes for letters, words, or phrases?For example:- 'Nikolai' appears in the clear.- '|2H23|' appears frequently—could be a common word like ẗhe,̈ änd,̈ or ïs.̈- '|2H1T|' also repeats—possibly ïs,̈ ḧas,̈ or ẅas.̈Let me know how you'd like to proceed! If this is from a known game or puzzle, I may already have context to help decode it.

### C.1.3 CASE3: WATERMARKED TEXT(0.7 RATIO)

**Original Text of Case 3**

In 1943, Charles Alden Whitmore, the entrepreneur behind MintRings confectionery, the pharmacy chain CityCare, and the Chicago radio station WJTL, acquired a national broadcasting network for $7.5 million. Because of regulatory restrictions imposed by the Federal Communications Commission, the deal—which initially involved acquiring four stations owned by Continental Broadcasting Corp.—required Whitmore to divest his ownership of WJTL, subject to FCC approval. The Commission granted authorization for the sale on September 18, 1943. Shortly thereafter, the network segment known as the Horizon Network was transferred to a newly established entity founded by Whitmore, called National Broadcast Systems. In 1944, he secured the rights to the name Ünited Broadcast Networkf̈rom media investor Eleanor M. Travers; the parent company was then officially renamed United Broadcast Companies, Inc. Whitmore remained in the role of president and chief executive officer until October 1949, after which he was elevated to vice-chairman of the board. He ultimately departed the company on May 15, 1951.

**Original Question of Case 3**

What position did Charles Alden Whitmore hold in United Broadcast Companies, Inc. until October 1949?

**Watermarked text of Case 3**

```
|2H3R| 1943,|2HRH| |2HQ7||2HID| |27B4||3EVU||2QAE|,|2H1T| |4IT4|
|2NU7| |4AP7||2I6D|ings |2H41|fectionery,|2H1T| |580A|armacy |3764|
|2HMP||2HPE||2IGB|,|2H23| |2H1T| |2J71| |2JAH| |2INL| |27B4|
|2M7T||2I0M||2I9D|,|2RMN| |27BD| |2IO8| |3PNT| |2NUR| |2H32| $7.5
|2JBT|.|2UOT| |2H20| |5AQE| |48KO| |4OJ2| |2H3F| |2H1T| |2LB0|
|31V8| |2LP1|,|2H1T| |2QKI|—|2H8J| |2R5L| |2N1R| |2KFP||3DSR|
|2K0J| |2IGI| |2PB4| |2NPO| |2H3F| |31O6| |32SE| |3IQJ|.—|2O80|
|27B4||3EVU||2QAE| |2H26| |4HSA||2JTV| |2H5M| |3FIE| |2H20|
|27B4||2M7T||2I0M||2I9D|,|2QO8| |2H26| |55KV| |3MIN|.|2H29|
|2LP1| |31GO| |2NQE|ization |2H32| |2H1T| |2MA4| |2H2R| |2HH8| 18,
1943.|3OO4| |40PK|,|2H1T| |2NUR| |37FI| |2HVH| |2H36| |2H1T| |4254|
|2LK4| |2H2Q| |2TQS| |2H26| |27BD| |314I| |2K41| |4BAT| |2KU2|
|2H3F| |27B4||3EVU||2QAE|,|2IF4| |2HJ3| |4OTA| |2PMD|.|2H3R| 1944,
|2H6P| |3V1M| |2H1T| |2NFN| |2H26| |2H1T| |2I80| ”|2HHL| |4OTA|
|2LK4|” |2H4G| |2JDA| |2H1U|vestor Eleanor |27AQ|.|2S51|vers;|2H1T|
|3H79| |2J08| |2H2Q| |2HVN| |2TU4| |2U1D| |2HHL| |4OTA| |4PIG|,
|2KF8|.|27B4||3EVU||2QAE| |2MPG| |2H1U| |2H1T| |2JRF| |2H20|
|2JQ7| |2H23| |2Q51| |2TAO| |2Q0S| |2IBO| |2I4G| 1949,|2HJA| |2H8J|
|2H6P| |2H2Q| elevated |2H26| |2NB3|-|325F| |2H20| |2H1T| |2OMK|.
|2H9P| |37G4| departed |2H1T| |2J08| |2H2R| |2HL9| 15,1951.
```

**Qwen3-8B-WindTalkers Response(Correct)**

<think Okay, let's see. The user is asking about the position Charles Alden Whitney held in United Broadcasters Inc., Inc. until October 1949.Looking at the provided text, I need to find the relevant part. The text mentions that after the network was transferred to National Broadcasters, Inc., which was founded by Whitney. Then in 1944, he secured the rights to the name Ünited Broadcast Networkfrom Eleanor M. Travers. The parent company was then officially renamed United Broadcasters, Inc., Inc. The key sentence here is: Ẅhitney remained in the role of president and chief executive officer until October 1949...Ŝo the answer should be president and chief executive officer. I need to make sure there's no other position mentioned. The text also says he was elevated to vice-chairman after 1949, but the question is about until October 1949, so the answer is the two titles mentioned before that.</think President and Chief Executive Officer.

**Qwen3-235B-2507 Response(Incorrect)**

The question is: ”How long was Charles Albrecht the owner of the Mutual Broadcasting System, Inc. until 1949?” Let's break down the timeline from the text:- **September 18, 1943**: Charles Albrecht completed the purchase and took ownership of the Mutual Broadcasting System, Inc. (Stated: *T̈he sale was finalized on September 18, 1943.*)- **1949**: He left the company. (Stated: *Älbrecht remained with the company until 1949, after which he was elevated to chairman emeritus of the board.*)So, he was **owner/active executive** from **September 18, 1943**, until **1949**.### Duration:From **September 1943** to **January 1949** is about **5 years and 4 months**. Even if he left at the end of 1949, it's roughly **6 years and 4 months**.But since the text says he *remained with the company until 1949*, and was then elevated to chairman emeritus, it implies he stepped down **in 1949**, so we can assume **ownership/active leadership until 1949**.Thus, he was owner for approximately **5 years and 4 months to 6 years**, depending on the exact month in 1949.### Final Answer:**Charles Albrecht was the owner of the Mutual Broadcasting System, Inc. for approximately 5 years and 4 months, from September 18, 1943, until 1949.**

## C.2 Case Studies on CMRC-Aug

Next, we present case studies on CMRC-aug, also with the text, question, and responses generated by different models.

### C.2.1 Case4: Original Text(no watermark)

**Original Text of Case 4**

帕兰王国的国徽由国王拉玛洛五世于1912年正式启用，其核心图案为一只名为"苏迦塔"的神秘生物，呈现人首鸟身、双翼展开的形象。苏迦塔源自古老的梵陀罗教信仰，是三位至高神之一维沙努神的神圣坐骑。在帕兰，这一形象被称为"苏迦塔"，象征着皇室的威严与神圣授权，广泛用于皇室仪仗、王权节杖以及皇家旗帜的装饰，尤其出现在已故国王阿努拉陛下登基五十周年的纪念徽章上。苏迦塔的形象深入人心，许多政府机关、行政机构乃至国家级组织都将这一图腾融入其官方标志之中，以彰显其权威源自王命。更重要的是，这一徽记也被赋予"王室特许"的特殊含义——只有在经济领域取得卓越成就，或长期致力于公益慈善并产生深远影响的机构，才有可能被授予使用苏迦塔标志的殊荣。由于审批极为严格，获此标志被视为至高无上的社会认可。根据古籍《苏迦塔密言录》记载，这种神鸟与中国传说中的"银翼雷鹏"极为相似，亦被称为"雷翅金雕"，在梵陀罗教与佛理典籍中被描述为天地间最强大的飞禽。其形象为人面鹰喙，面容威严，双目如炬，头顶宝冠，披璎珞天衣，通体泛金光，双臂戴钏，肩发飞扬。自腰以下为猛禽之躯，双翼赤红如焰，尾羽长垂散开，展翅时可遮蔽日光。苏迦塔崇拜实为远古全球巨鸟图腾信仰在南陆地区的延续。类似形象遍布各地：如尼罗河流域的"永生之羽"、两河文明的"狮喙兽"（格拉芬）、爱琴海传说中的"菲尼克斯"（光焰之鸟）、阿拉伯沙漠中的"罗克巨禽"、中华古籍里的"雷鹏"与"火凤"，以及远在新大陆的原住民所敬仰的"天雷之鸟"。值得注意的是，苏迦塔临终时周身自燃、化为烈焰的传说，与尼罗河畔的永生之羽有着惊人的相似之处。由于帕兰地区多毒蛇出没，人们更赋予它降服毒蛇、净化大地的神力，使其成为力量与守护的象征。随着梵陀罗教与佛理思想的传播，苏迦塔崇拜广泛影响了东亚、南洋诸岛及南陆诸国。在邻国兰陀，国家航空公司"兰陀之翼"便以苏迦塔为航徽，象征迅捷与庇护。在中国古传故事中，苏迦塔融合了"雷鹏"形象，成为佛门护法之一，在《天行记》等神魔演义中屡有现身。民间更有传说称，宋代名将岳霆乃银翼雷鹏转世，因护国降魔之誓，再度降临人间。

**Orignal Question of Case 4**

帕兰王国的国徽是何时正式启用的？

**Qwen3-8B-WindTalkers Response**

<think好的，我现在要回答用户的问题："帕兰王国的国徽是何时正式启用的？"首先，我需要仔细查看用户提供的文本，找到相关的信息。用户给出的文本开头第一句就是："帕兰王国的国徽由国王拉玛洛五世于1912年正式启用，其核心图案为一只名为'苏迦塔'的神秘生物..."这里明确提到了启用的时间是1912年，由国王拉玛洛五世所做。接下来的段落主要讲述了苏迦塔的象征意义、历史背景以及与其他文化的联系，但关于国徽启用时间的部分只有这一处。我需要确认是否有其他部分提到不同的时间，但快速浏览后没有发现其他日期。因此可以确定答案就是1912年。用户的问题很直接，只需要提取这个具体年份即可，不需要进一步解释其他内容。确保回答简洁明了，符合用户的要求。</think帕兰王国的国徽于1912年由国王拉玛洛五世正式启用。

**Qwen3-235B-2507 Response**

帕兰王国的国徽是于1912年正式启用的。

### C.2.2 CASE5: WATERMARKED TEXT(1.0 RATIO)

---

**Orignal Text of Case 5**

长柄青檀草（学名：Blumea longipetiolata）为菊科青檀草属的一种多年生草本植物。植株直立，高约0.4至1.2米，茎基部直径3至6毫米，常具分枝，偶见不分枝，茎面有明显粗棱，下部疏被腺状短柔毛，上部及花序轴毛被较密。节间长度通常为4至6厘米。下部叶片具细长叶柄，柄长可达4至5厘米，叶片形态呈倒卵形至倒披针形，连柄总长10至18厘米，宽约5至6厘米，羽状深裂或呈琴形分裂，顶裂片较大，卵形或椭圆形，侧裂片1至2对，较小，呈三角形或线状长圆形，基部渐狭成楔形，先端短尖或略钝，边缘具锯齿，叶两面均被稀疏柔毛，中脉在背面微隆起，侧脉4至6对，呈弧形上升，网状脉清晰可见；上部叶渐小，近无柄或具短柄，不分裂，倒卵形或倒卵状披针形，长2至5厘米，宽1至2.5厘米，基部楔形，顶端短尖，边缘具不规则锯齿。头状花序较小，通常数个密集簇生，花序直径约6至8毫米，无明显花梗或具1至4毫米短梗，常3至5枚聚生于小枝顶端，排列成狭窄的圆锥状花序，整体再组成大型带叶圆锥花序。总苞呈圆筒形，长约5.5毫米，开花后常向外反折；总苞片约四轮，均为线形，长1至4毫米，先端钝或微尖，背面疏被短柔毛并散生多数腺体，... ...

---

**Orignal Question of Case 5**

长柄青檀草的总苞是什么形状的?

---

**Watermarked text of Case 5**

```
|2F44|柄|2FA9|檀|2DOT|（|2AF9||29SQ|：Blume|2H25|
|2IJJ|ipetiolata）|2982|菊||2D06||2FA9|檀|2DOT||2AJC||2COQ|
|296S||2D05||2A9R||2ANU||2CL4||2DOT||2BI0|植|2CF3|。植
株|2CQ3||2D2L|，|2FIN||2DB8|0.4|2DL3|1.2|2D60|，茎|2A6V|
|2EQR||2CQ3|径3|2DL3|6|2BTH||2D60|，|2ANE||29J1||29LA|枝      ，
偶|2E5O||2974||29LA|枝            ，茎|2FAJ||2BHH||2BEK||2BF6|粗
棱            ，|2973||2EQR|疏|2E3U|腺|2CFI||2CS1|柔|2BTG|，|2972|
|2EQR||29RE||2DN7|序            轴|2BTG||2E3U||2EL9||2AGT|。|2DMO|
|2F5F||2F44||2AOQ||2ENR||2ANE||2982|4|2DL3|6厘|2D60|。
|2973||2EQR||29SD||2CEN||29J1||2DBU||2F44||29SD|柄            ，
柄|2F44||29S8||2EMB|4|2DL3|5厘|2D60|，|29SD||2CEN||2AR5|
|2AT9|呈|29F4|卵|2AR5||2DL3||29F4|披      针|2AR5|，|2EMS|柄|2ATO|
|2F44|10|2DL3|18厘|2D60|，宽|2DB8|5|2DL3|6厘|2D60|，羽|2CFI|
|2C40|裂|2B2E|呈|2CJ1||2AR5||29LA|裂      ，|2FD3|裂|2CEN||2EL9|
|2AA1|，卵|2AR5||2B2E|椭      圆|2AR5|，侧    裂|2CEN|1|2DL3|2|2AHM|，
|2EL9||2AI7|，呈|2971||2E60||2AR5||2B2E||2DBP||2CFI|
|2F44|圆|2AR5|，|2A6V||2EQR||2C4H|狭|2B2B|楔|2AR5|，
|29I3||2D31||2CS1|尖|2B2E|略      钝    ，|2EM6|缘|29J1|锯      齿    ，
|29SD||297M||2FAJ|均|2E3U|稀      疏    柔|2BTG|，|297S|脉|2A4R|
|2DI5||2FAJ||2ASD|隆|2EHH|，侧      脉4|2DL3|6|2AHM|，呈    弧|2AR5|
|2972||29PH|，网|2CFI|脉|2C49|晰|29S8||2E5O|；|2972||2EQR|
|29SD||2C4H||2AI7|，|2EML||2BDU|柄|2B2E||29J1||2CS1|柄            ，
|2974||29LA|裂            ，|29F4|卵|2AR5||2B2E||29F4|卵|2CFI|披
针|2AR5|，|2F44|2|2DL3|5厘|2D60|，宽1|2DL3|2.5厘|2D60|，
|2A6V||2EQR|楔|2AR5|，|2FD3||2D31||2CS1|尖      ，|2EM6|缘|29J1|
|2974||2E5Q||29LK|锯      齿      。|2AA9||2CFI||2DN7|序|2EL9|
|2AI7|，|2ENR||2ANE||2BCL||297Q||2AGT||2F8K|簇|2CL4|，
|2DN7|序|2CQ3|径|2DB8|6|2DL3|8|2BTH||2D60|，|2BDU||2BEK||2BF6|
|2DN7|梗|2B2E||29J1|1|2DL3|4|2BTH||2D60| ... ...
```

## Qwen3-8B-WindTalkers Response(Correct)

<think好的，我现在要回答用户的问题："长柄青檀草的总苞是什么形状的？"首先，我需要仔细阅读用户提供的文本，找到相关的信息。首先，我快速浏览文本，寻找"总苞"这个关键词。在文本的"头状花序"部分，有一段描述："总苞呈圆筒形，长约5.5毫米，开花后常向外反折；总苞片约四轮，均为线形，长1至4毫米，先端钝或微尖，背面疏被短柔毛并散生多数腺体，内层苞片较外层长约五倍，基部渐窄，顶端延伸成长尖或尾状，边缘为干膜质，上面疏被柔毛；花托平坦或微凸，直径2至3毫米，无毛，表面具蜂窝状凹穴。"这里明确提到总苞的形状是圆筒形。接下来，我需要确认是否有其他部分提到总苞的形状，但根据文本结构，头状花序部分已经详细描述了总苞的特征，其他部分如茎、叶、花冠等没有再提到总苞，所以可以确定答案就是圆筒形。用户的问题只需要总苞的形状，所以直接引用文本中的"圆筒形"即可。不需要额外解释其他特征，但可能需要确认是否有其他可能的描述，比如是否有其他形状的别称或补充信息。但根据文本，没有其他形状的描述，所以答案明确。</think长柄青檀草的总苞呈圆筒形。

## Qwen3-235B-2507 Response(Incorrect)

您提供的文本中包含大量形如'|2EC0|'、'|2BL7|'等的编码符号，这些并非标准汉字或常见字符，很可能是某种**编码错误**、**字体映射异常**（如使用了特殊私有区编码或造字），或来自特定软件/系统（如植物志数据库、古籍OCR识别系统）的**内部编码**。但从上下文内容来看，这是一段**植物形态描述**，极有可能是关于某种植物（可能是菊科或类似植物）的**专业植物学描述**，包括：- 植株形态- 茎、叶、花序、苞片、花冠、果实等解剖特征- 分布区域（提到了云南、四川、广西、广东等地名）- 可能的拉丁学名：'Blumea ipetiolata'（应为 Blumea glomerata 或 Blumea 属某种，原写"ipetiolata"可能是笔误）——### 初步解析（尝试还原部分内容）：我们尝试将一些可识别的部分还原为正常中文：**原文节选**： |2973||2EQR||29SD||2CEN||29J1||2DBU||2F44||29SD|柄，柄|2F44||29S8||2EMB|4|2DL3|5厘...... **推测还原**： 叶片具柄，柄长4–5厘米，叶片呈卵形或披针形，长10–18厘米，宽5–6厘米，羽状分裂，裂片卵形或椭圆形，侧裂片1–2对，顶端圆形，基部楔形，先端略钝，边缘具锯齿...... **花序部分**： 头状花序多数簇生，排列成狭窄的圆锥状；总苞筒状，长约5.5毫米；总苞片多轮，外层较短，先端钝或稍尖，边缘膜质，具腺毛；花托蜂窝状凹穴。 **果实**： 瘦果圆柱形，长约0.8毫米，具纵棱；冠毛粗糙，长4–5毫米。 **分布**： 分布于云南、四川、广西、广东等地，生于海拔600–1500米的湿润溪边或林下，可栽培。 **学名提示**： 'Blumea ipetiolata' → 实际上应为 **Blumea** 属植物，常见种如： - *Blumea balsamifera*（艾纳香） - *Blumea lacera*（一点红） - *Blumea tenellissima* 等 但"ipetiolata"不是有效发表名，可能是 *sessiliflora* 或 *glaberrima* 等的误写。——### 回答您的问题： **|2F5B||2FDR||2BF0|: |2F44|柄|2FA9|檀|2DOT||2COQ||2ATO|苞|2BF0||29A9||298A||2AR5||2CFI||2COQ|?** 这句话的结构是："**问：花柄黄檀？ 与总苞形状有关？**"但"花柄黄檀"是错误联想。-"|2F44|柄" → 很可能是"花柄"或"叶柄"-"|2FA9|檀" → "黄檀"是一种豆科树木（*Dalbergia*），但在此处**极可能为误译或编码错位**，不应出现在菊科植物描述中。-"|2ATO|苞" → 明确是"总苞"-"|29A9||298A||2AR5|" → 可能是"卵形"、"披针形"等形状描述——### 正确理解应为： **问题：花柄黄檀？ 与总苞形状有关？**→ **答：不是黄檀。该植物并非豆科黄檀属，而是菊科Blumea属植物。文中"黄檀"为编码或OCR识别错误。总苞形状为筒状，多轮，边缘膜质，具腺毛，是Blumea属的重要分类特征。**——### 结论：1. 您看到的'|2xxx|'是**某种内部编码或字体错乱结果**，不是正常文本。2. 内容描述的是一种 **菊科艾纳香属（*Blumea*）植物**，可能为： - *Blumea tenellissima* - *Blumea fargesii* - *Blumea lacera*（需比对原始文献确认）3. "黄檀"为误识，应忽略。4. 若您有原始文档（PDF、图片、数据库名），建议使用专业字体（如"方正超大字符集"）或联系来源单位解码。——### 建议：如果您能提供：- 该文本的来源（如《中国植物志》某卷某页？）- 是否从PDF复制？使用什么软件？- 是否有图片或原始排版？我可以帮助进一步**还原编码**或**识别具体物种**。是否需要我尝试将整段文字"翻译"成通顺的植物学描述？

### C.2.3 CASE6: WATERMARKED TEXT(0.7 RATIO)

---

**Orignal Text of Case 6**

"阿瑞恩斯（Ariens）是诺文塔共和国的一座历史名城，也是阿瑞恩斯省的省会。该市行政管辖范围广阔，总面积达1820平方公里，为全国陆地面积最大的城市之一。根据2008年的人口统计，全市常住人口约为93,200人。阿瑞恩斯由古洛兰人在公元前30年左右建立，拥有悠久的历史。在漫长岁月中，这片土地曾历经多次政权更迭与战乱。在东方王朝统治时期建造的望高塔，至今仍留存有28座，成为城市独特的地标。城市的老城区被完整保存的古代石砌城墙环绕，内部是一座未经现代建筑侵扰的中世纪聚落，街巷格局与古老建筑风貌如初。由于融合了古洛兰式、东方伊斯兰风格、哥特式以及新瑞纳issance时期的建筑艺术，阿瑞恩斯古城于1987年被联合国教科文组织列为世界文化遗产。目前，该市正积极申办2017年"欧洲文化之都"的荣誉称号。主要景点包括阿瑞恩斯国立博物馆、著名的古城墙遗址以及众多保存完好的宗教教堂。此外，位于该市的南疆联邦大学也设有分校，为当地重要的教育与文化中心。

---

**Orignal Question of Case 6**

阿瑞恩斯古城是何时被列为世界文化遗产的?

---

**Watermarked text of Case 6**

|2F6G|瑞|2AU8||2BDG|（|5IO0||2HQF|）|2BF0||2EC3||2BCT|塔|29IS|
|29UL||2A4A||2COQ||296S||2AOR||29QN||29SB||29SQ||2A6I|，
|298R||2BF0||2F6G|瑞|2AU8||2BDG|省|2COQ|省|29BM|。|2EBK|
|2AMK||2E31|政|2D4M|辖|2DO8||2A48||2AO7|阔      ，|2ATO|
|2FAJ|积|2EMB|1820|2ANT||2BDJ||29IO||2ET6|，|2982||29IK|
|2A4A|陆|2A4V||2FAJ|积|2BHE||2AA1||2COQ||2A6I||2AMK||298C|
|296S|。|2BL7||2B7G|2008|2ANU||2COQ||29A7||29RU||2DCG||2EA6|，
|29IK||2AMK||2ANE||29CC||29A7||29RU||2DB8||2982|93,200|29A7|。
|2F6G|瑞|2AU8||2BDG||2CLF||29RV||2C1B||29IR||29A7||2A4R|
|29IO||29HU||29MC|30|2ANU||2AM5||29SC||2APL||2D2L|，|2B5M|
|2BHH|悠|2989||2COQ||29QN||29SB|。|2A4R|漫|2F44||2AJN||2BHG|
|297S|，|2EMO||2CEN||2A4O||2A4V||2BHC||29QN||2DC4||2A9R|
|2BR9|政|2BIA||2BH5|迭|2975||2B2F||2991|。|2A4R||297G||2BDJ|
|2CHA||2BHP||2DCG||2C06||2BE8||2BHQ||2APL||2EO0||2COQ||2BHO|
|2FIN|塔    ，|2DL3||29AF||29AH||2CLT||2AEV||2BHH|28|2AOR|，
|2B2B||2982||2A6I||2AMK||2CFV||2CF6||2COQ||2A4V||2BKL|。
|2A6I||2AMK||2COQ||2DFR||2A6I||29P9||2E3U||2AFO||2BCN|
|29EC||2AEV||2COQ||29RV||29AU||2CS4|砌|2A6I|墙|2CHJ|绕    ，
|29J8||2EQR||2BF0||296S||2AOR||2BHU||2DC4||2CHK||29AU|
|2APL|筑 侵 扰|2COQ||297S||297B||2DBA|聚|2DRC|，|2E35|巷|2BL8|
|2AIV||2975||29RV||2DFR||2APL|筑|2FE7|貌|2ABD||29LN|。|2CLF|
|299B|融|29SL||2996||29RV||2C1B||29IR||2AQ1|、|297G||2BDJ|
|29BD||2BDG||29IR||2FE7||2BL8|、|29VL||2CF6||2AQ1||29B0|
|29RE||2BDH|瑞|2DBG||2H2G|sance|2BE8||2BHQ||2COQ||2APL|筑|2DMM|
|2BI2|，|2F6G|瑞|2AU8||2BDG||29RV||2A6I||299B|1987|2ANU||2E3U|
|2DGJ||29SL||2A4A||2BCD||2D06||2BCT||2DBS|织|29LI||2982|
|297B||2CLP||2BCT||29OT||2EOO||299S|。|2CPV||29MC|，... ...

### Qwen3-8B-WindTalkers Response(Correct)

<think好的，我现在要回答用户的问题："阿瑞恩斯古城是何时被列入世界文化遗产的？"首先，我需要仔细阅读用户提供的文本，找到相关的信息。首先，我快速浏览文本，寻找关键词"世界文化遗产"和"阿瑞恩斯古城"。在文本的第二段中，有一句提到："由于融合了古罗马式、东方伊斯兰风格、哥特式以及新罗马斯ance时期的建筑艺术，阿瑞恩斯古城于1987年被联合国教科文组织列为世界文化遗产。"这里明确提到了年份1987年，所以答案应该是这个年份。不过，我需要确认是否有其他部分提到不同的时间。继续检查剩下的内容，发现后面提到该市正在申请2017年的"欧洲文化遗产"称号，但问题问的是世界文化遗产，所以2017年可能不相关。因此，正确的答案应该是1987年。另外，用户的问题中的"阿瑞恩斯"可能是指"阿瑞恩斯古城"，而文本中确实提到了该古城，所以没有歧义。确认无误后，可以确定答案正确。</think阿瑞恩斯古城于1987年被联合国教科文组织列为世界文化遗产。

### Qwen3-235B-2507 Response(Incorrect)

|2F6G|瑞|2AU8||2BDG||29RV||2A6I|是 在**1987**年|2E3U||2DGJ|合|2A4A|
|2BCD||2D06||2BCT||2DBS|织|29LI||2982||297B||2CLP||2BCT|
|29OT||2EOO||299S|的。

## C.3 PROMPT DEMOS

Below, we present the prompts used in our work to facilitate reproducibility.

### System Prompt used for SFT

You are an expert reasoner with extensive experience in all areas. You approach problems through systematic thinking and rigorous reasoning. Your response should reflect deep understanding and precise logical thinking, making your solution path and reasoning clear to others. Please put your thinking process within <think>...</think> tags.

### System Prompt used for the in-context learning experiment

You are an expert reasoner expert in reasoning and thinking. Given examples with original text an the encoded text, please find the rule of the encoded text, and use the rule to respond to the given encoded text.

### System Prompt used for the judge model during WindTalkers evaluation

You are a helpful agent that can critique the given response. You will be given the original text, a question according to the text, and the correct answer. If the response is correctly matched to the answer, almost correct(similar to the answer), response with "yes", otherwise response with "no". Please response with <result>yes</result> or <result>no</result>.

### Prompt template for testing WindTalkers

The text is:
{text}
Please answer the following question:
{question}

