# OpenReview forum: "Windtalkers: Watermarking Open-Source LLMs with Ciphered-Instruction"
_ICLR.cc/2026/Conference — ICLR 2026 Conference Withdrawn Submission_

### Official Review · Reviewer_sbu2 · 2025-10-27

**Soundness:** 2
**Presentation:** 1
**Contribution:** 1
**Rating:** 2
**Confidence:** 4

**Summary:**

This paper proposes WindTalkers, a method for planting a fingerprint into an LLM, which enables checking if a given LLM is equal to the original LLM, or a post-trained version. This enables ownership claims/tracking of open-source LLMs.

The method is to train the model to respond to cipher-encoded instructions. Therefore, if a given LLM is able to respond to the ciphered instructions, it can be detected as from the original model. The paper runs experiments showing that the ability to understand ciphered instructions remains after additional post-training.

**Strengths:**

The problem of detecting whether a model was copied/stolen or fine-tuned from another model is important given today’s competitive LLM landscape. The cipher method proposed in the paper is simple to understand.

**Weaknesses:**

## Difference between watermarking of text (LLM outputs) versus watermarking/fingerprinting of model itself

There are two types of watermarking in LLMs that have very different goals:

1.  **Watermarking of text (LLM outputs):** The goal here is to answer the question: *“Is this text LLM-generated?”* To do so, a hidden statistical pattern is embedded into all texts generated by the LLM. Then, this pattern enables the detection of LLM-generated text. Some papers on this topic (which were cited in the paper) include [Kirchenbauer et al. (2023)](https://proceedings.mlr.press/v202/kirchenbauer23a.html), [Kuditipudi et al. (2023)](https://arxiv.org/abs/2307.15593), [Christ et al. (2024)](https://proceedings.mlr.press/v247/christ24a.html), [Hu et al. (2023)](https://arxiv.org/abs/2310.10669), and [Xu et al. (2024a)](https://arxiv.org/abs/2403.10553).
2. **Watermarking/fingerprinting of the model itself:** The question here is: *“Is this model copied/stolen/fine-tuned from my model?”* This is the setting that the method proposed in this paper addresses. [Xu et al. (2024b)](https://arxiv.org/abs/2401.12255) addresses this setting, and also has a good discussion about the difference between text watermarking and model fingerprinting (see Section 2.3).

This paper (the submission) does not make a distinction between these settings. Papers from both settings are cited and mixed together in the introduction, and the setting itself is never clearly stated. This makes it difficult and confusing to understand what the problem formulation is until much later in the paper.

The abstract mentions the ineffectiveness of generation-time watermarks. However, generation-time watermarks are only applicable to text watermarking, where we want to insert the watermark into all outputs at generation time. For model fingerprinting, the watermark/fingerprinting must already be planted into the model, as we are running detection on the model itself, not its text outputs.

## Lack of novelty, missing many important references

There are many works on model watermarking/fingerprinting that this paper does not mention. For example (not an exhaustive list), [Xu et al. (2024b)](https://arxiv.org/abs/2401.12255), [Nasery et al. (2025)](https://arxiv.org/abs/2502.07760), [Gloaguen et al. (2025a)](https://arxiv.org/abs/2505.16723), [Gu et al. (2022)](https://arxiv.org/abs/2210.07543), [Li et al. (2023)](https://ojs.aaai.org/index.php/AAAI/article/view/26750), [Russinovich & Salem (2024)](https://arxiv.org/abs/2407.10887), and [Zeng et al. (2024)](https://proceedings.neurips.cc/paper_files/paper/2024/hash/e46fc33e80e9fa2febcdb058fba4beca-Abstract-Conference.html).

In particular, this paper is very similar to [Xu et al. (2024b)](https://arxiv.org/abs/2401.12255), [Nasery et al. (2025)](https://arxiv.org/abs/2502.07760), and [Gloaguen et al. (2025a)](https://arxiv.org/abs/2505.16723). [Xu et al. (2024b)](https://arxiv.org/abs/2401.12255) and [Nasery et al. (2025)](https://arxiv.org/abs/2502.07760) fingerprint an LLM by training it to learn specific secret (input, output) pairs. Then, detection is seeing if a model outputs the desired response for the secret inputs. They find that this fingerprinting method is robust to further fine-tuning and minimally affects model performance. [Gloaguen et al. (2025a)](https://arxiv.org/abs/2505.16723) fingerprints by training with entire semantic domains as the fingerprint queries, instead of specific inputs, and with statistical patterns as the fingerprint output, instead of specifically memorized outputs.

The omission of these closely related works is a significant concern in my opinion. A widely used terminology for this problem setting is **fingerprinting** (as in the aforementioned works), yet this word does not appear anywhere in this paper.

## No baseline comparisons in experiments

There are no comparisons to other existing methods except for one comparison with Double-I-Gen in Figure 4, making it difficult to evaluate and contextualize the proposed WindTalkers method.

**Questions:**

1. Line 44: the introduction states that there are three categories of watermarks: generation-time watermarks, model-embedded watermarks, and post-hoc watermarks. However, no citations or definitions are provided. Could you clarify what these categories are? In particular post-hoc watermarks, as they are not mentioned anywhere else in the paper.
2. The paper mentions [Gloaguen et al. (2025b)](https://arxiv.org/abs/2502.10525) as a backdoor-based method (lines 130, 136), but the cited paper does not seem to mention backdoors at all.
3. Line 205 states that only the 1,000 most common Chinese characters and 10,000 most common English words are used. Does this mean that only these words are encoded into base-36, and other words are left as is?
   1. If so, what does a watermark ratio of 100% mean? All tokens are encoded, or only all tokens that fall into the most common subset?
4. Line 207 discusses encoding at word level for English and character level for Chinese. However, aren’t tokens the unit of encoding, so the level of encoding should be pre-determined by the tokenizer?
5. Line 238 uses the bert-base-multilingual-cased tokenizer. Why not use the native tokenizer for the LLM being trained?
6. Table 1: what is the difference between the Base and No-Watermark models?
   1. Why is there such a difference in AIME scores between Base and No-Watermark? InternLM2.5-7B-Base gets 0 on AIME, but InternLM2.5-7B-No-Watermark gets 73.3.
7. Table 1: Why does InternLM2.5-7B-WindTalkers perform significantly better on original CMRC-Aug than InternLM2.5-7B-Base (97 vs 83)?
8. Line 298 states: *“The fine-tuned models, Qwen38B-WINDTALKERS and InternLM2.5-7B-WINDTALKERS, effectively close the performance gap between original and watermarked data.”* However, the gap seems quite large still. On SQuAD-Aug, the score on the original data is above 90, while the score on the watermarked data is below 40\.

---

### Official Review · Reviewer_cP32 · 2025-10-28

**Soundness:** 2
**Presentation:** 2
**Contribution:** 2
**Rating:** 4
**Confidence:** 4

**Summary:**

This paper proposes a method for embedding a robust and detectable ownership signal into open-source LLMs. The key idea is to train models to understand ciphered instructions, which are inputs encoded through a Base36 transformation of token IDs with an offset without affecting their normal generation quality. During inference, the model’s ability to correctly interpret these ciphered inputs serves as evidence of watermark presence.

Unlike generation-time watermarking methods that modify output distributions, or backdoor-based watermarking that injects visible behavioral triggers, WindTalkers embeds the watermark directly into the instruction-following layer of the model. The ciphered-instruction learning is framed as a low-resource language adaptation task, which makes the watermark both semantically invisible and resilient to subsequent fine-tuning or reinforcement learning. The authors demonstrate robustness under several post-training operations (SFT, GRPO, and “defensive fine-tuning”), as well as resistance to algorithm-aware but key-unknown adversaries.

**Strengths:**

1. The paper tackles a critical unsolved problem: embedding ownership verification signals in open-weight LLMs where full parameter access invalidates most prior watermarking schemes.

2. The authors systematically evaluate the persistence of the watermark under SFT, RL (GRPO), and defensive post-training. The empirical results demonstrate that the encoded instruction space remains stable under diverse optimization regimes, which is a strong sign that the embedding lies in a robust submanifold of the model’s parameter space.

3. The work contributes meaningfully to the emerging area of IP protection and provenance verification for open-weight models.

**Weaknesses:**

1. I believe WindTalkers should be classified as fingerprinting rather than watermarking.  The detection relies on whether the model can interpret ciphered instructions defined by an offset parameter $o$, and different offsets produce distinct model variants. The verification is deterministic (binary), matching the notion of model fingerprinting where each model instance carries a unique identifier, rather than a shared statistical watermark signal. The authors should clarify this distinction, as it affects both the conceptual framing and the choice of baselines.

2. Although the proposed method is novel, it appears to be an unnecessarily complex way to solve a problem that can be addressed using simpler techniques. Prior studies, such as "Instructional Fingerprinting of Large Language Models" (NAACL 2024) and "Instructions as Backdoors" (NAACL 2024), have already shown that lightweight instruction-tuning or small backdoor datasets can embed stable and detectable fingerprints. These simpler approaches achieve comparable robustness without requiring the design of an additional “minor language.” Therefore, the practical advantage of WindTalkers over existing fingerprinting or backdoor methods remains unclear. The authors should include direct comparisons to these baselines and quantify the trade-off between complexity, computational cost, and watermark robustness.

3. The recent work by Yijie Xu et al. (2025), “Mark Your LLM,” further demonstrates that simple fingerprinting methods can achieve the same three properties claimed as the main contributions of WindTalkers:
   (1) stable detectability,
   (2) minimal interference with normal language tasks, and
   (3) persistence across fine-tuning and reinforcement learning.
   This evidence suggests that WindTalkers’ advantages might not stem from conceptual innovation but rather from implementation details. The authors should discuss these related works directly and clarify whether WindTalkers offers a fundamentally new mechanism or primarily an engineering refinement.

4. The paper would be improved by explicitly citing and comparing against the following works to better situate its contributions and avoid conceptual overlap:
   - Xu, J., Wang, F., Ma, M. D., Koh, P. W., Xiao, C., & Chen, M. (2024). Instructional Fingerprinting of Large Language Models. NAACL 2024.
   - Xu, J., Ma, M. D., Wang, F., Xiao, C., & Chen, M. (2024). Instructions as Backdoors: Backdoor Vulnerabilities of Instruction Tuning for Large Language Models. NAACL 2024.
   - Xu, Y., Liu, A., Hu, X., Wen, L., & Xiong, H. (2025). Mark Your LLM: Detecting the Misuse of Open-Source Large Language Models via Watermarking. ICLR 2025 Workshop on GenAI Watermarking.
   - Gu, C., Li, X. L., Liang, P., & Hashimoto, T. (2024). On the Learnability of Watermarks for Language Models. ICLR 2024.


5. Furthermore, this paper seems not discussed the usage of LLMs, which is required by ICLR this year.

**Questions:**

See weaknesses

---

### Official Review · Reviewer_75qh · 2025-10-30

**Soundness:** 3
**Presentation:** 3
**Contribution:** 2
**Rating:** 4
**Confidence:** 4

**Summary:**

This paper proposes WindTalkers, a robust watermarking method for open-source LLMs that embeds a detectable signature directly into model weights through gradient-based fine-tuning. The core innovation involves transforming training instructions using a cryptographic encoding scheme, where token IDs are shifted by a secret offset and converted to base-36 representations, creating a minor language intelligible only to the watermarked model.

**Strengths:**

- WindTalkers demonstrates strong robustness against post-training techniques like SFT and RL, maintaining watermark integrity.
- This approach ensures high detectability where non-watermarked models fail to respond coherently to encoded prompts—while preserving fidelity on standard tasks, as only instruction processing is altered.

**Weaknesses:**

- The paper's critique of backdoor-based watermarking methods appears overstated. The authors claim these methods significantly alter model output, but this effect is only triggered under specific conditions that are unlikely to occur during normal usage. For regular users, these backdoors remain dormant and don't affect performance, making the criticism somewhat exaggerated.
- Furthermore, WindTalkers itself can be viewed as essentially a backdoor-based approach, differing primarily in its trigger mechanism. The method uses base-36 encoded tokens as triggers - a relatively minor modification compared to existing techniques. While the authors acknowledge this is a "trigger-based watermarking technique" in Section 3.1, their experimental comparisons with prior trigger-based methods remain limited. The paper would benefit from more comprehensive comparisons with existing backdoor-based approaches in the experiments.
- The current comparison with Double-I watermark provides some insights, but a broader comparison against other trigger-based methods would strengthen the claims about WindTalkers' novelty and effectiveness. The simplicity of the base-36 encoding, while innovative, raises questions about whether this represents a fundamental advancement or merely an incremental improvement over existing trigger-based paradigms.

**Questions:**

- You demonstrate robustness to SFT/RL, but what about other common post-training operations like pruning, quantization, or knowledge distillation? Could these remove the watermark?
- You compared with Double-I watermark, but why not include other recent backdoor-based methods for a more comprehensive comparison?

---

### Official Review · Reviewer_d4Ts · 2025-10-31

**Soundness:** 2
**Presentation:** 2
**Contribution:** 2
**Rating:** 2
**Confidence:** 4

**Summary:**

This paper proposes WindTalkers, a novel watermarking method for open-source large language models (LLMs) that encodes training instructions with a cipher-like base-36 transformation. The approach aims to implant a watermark that can survive common post-training modifications (e.g., SFT, RL) without degrading model performance. The key idea is that a watermarked model can decode ciphered instructions, while an unwatermarked one cannot. The authors present experiments on Qwen3 and InternLM models and claim robustness and fidelity against fine-tuning and RL training.

**Strengths:**

- The paper targets an important problem: how to watermark open-source LLMs whose generation processes can be freely modified by users.
- Framing watermarking as learning an auxiliary “ciphered language” can be interesting.

**Weaknesses:**

- The authors do not measure or discuss whether learning to decode ciphered instructions affects the model’s general capabilities or efficiency (e.g., perplexity, reasoning accuracy, or transfer performance on standard benchmarks). A proper trade-off study is missing.
- The “cipher” is a deterministic base-36 encoding with a fixed offset; the security argument is largely conceptual and untested against active attacks (e.g., cipher inversion or data re-encoding).
- The proposed “Detection Score” (ratio of accuracies) is unconventional and difficult to interpret relative to standard detection metrics such as AUC or TPR@FPR.
- The paper lacks any theoretical justification for why the learned cipher capability should remain stable after post-training, or under what conditions the watermark is provably detectable or irrecoverable. There is no information-theoretic or optimization-based formulation supporting the claimed robustness and security.
- The evaluation omits comparison with other generative or model-embedded watermarking baselines.

**Questions:**

- How does WindTalkers compare quantitatively with established watermarking baselines?
- Could the authors provide detailed analyses of how watermarking affects general performance such as reasoning accuracy, fluency, and efficiency on diverse benchmarks?

Please refer to the weakness part.

---

### Note · Authors · 2025-11-12

**Comment:**

We sincerely thank the reviewers for their efforts during the review process!

Regarding the concern raised by Reviewer **sbu2**, we find that our proposed method is indeed similar to existing LLM fingerprinting techniques. Consequently, the omission of these LLM fingerprinting methods in our related work section and experiments section is indeed an oversight. Therefore, we have decided to withdraw the paper from consideration.

Sincerely,

The Authors

**Withdrawal Confirmation:**

I have read and agree with the venue's withdrawal policy on behalf of myself and my co-authors.